# An Empirical Orthogonal Function Study of the Ionospheric TEC Predicted Using the TIEGCM Model over the South Atlantic Anomaly in 2002 and 2008

**Jing Yu** [1,2], **Zheng Li** [3,*], **Yan Wang** [3], **Jingjing Shao** [1], **Luyao Wang** [1], **Jingyuan Li** [3], **Hua Zhang** [3], **Xiaojun Xu** [2] **and Chunli Gu** [4]

1   School of Mathematics and Statistics, Nanjing University of Information Science and Technology, Nanjing 210044, China
2   State Key Laboratory of Lunar and Planetary Science, Macau University of Science and Technology, Macau 999078, China
3   Institute of Space Weather, Nanjing University of Information Science and Technology, Nanjing 210044, China
4   Beijing Institute of Applied Meteorology, Beijing 100081, China
*   Correspondence: zli@nuist.edu.cn

**Abstract:** In this study, the variability of the ionospheric total electron content (TEC) in the South Atlantic Anomaly (SAA) in the solar maximum of 2002 and the solar minimum of 2008 were compared by using an empirical orthogonal function (EOF) analysis. The ionospheric TEC data were simulated using the National Center for Atmospheric Research Thermosphere-Ionosphere-Electrodynamics General Circulation Model (TIEGCM). The first three EOFs accounted for 94.8% and 93.86% of the variability in the data in 2002 and 2008, respectively. The results showed that the TEC variations of the first three EOFs were generally consistent in 2002 and 2008. The first mode showed the equatorial anomaly caused by plasma drift and the east–west asymmetry possibly caused by the change in geomagnetic declination and zonal wind; $EOF_2$ exhibited the zonal variation influenced by the solar EUV radiation and the semiannual variation possibly controlled by the $[O/N_2]$, solar zenith angle, and atmospheric circulation. $EOF_3$ suggested an equatorial anomaly and winter anomaly influenced by the $[O/N_2]$ variation. However, the values and amplitude variations in the TEC were significantly greater in the solar maximum than that in the solar minimum, and the spring–autumn asymmetry of the TEC was more obvious in the solar minimum. In addition, we used the EOF method to extract the annual variation characteristics of the time coefficients and carried out a correlation analysis. The results showed that the annual variation in the TEC in 2002 was mainly affected by the solar EUV radiation, which was strongly correlated with F10.7 (r = 0.7348). In contrast, the TEC was mainly influenced by the geomagnetic activity in 2008 and had a strong correlation with Dst (r = −0.7898).

**Keywords:** South Atlantic Anomaly; ionospheric total electron content; empirical orthogonal function

## 1. Introduction

The ionosphere is an important part of the solar–terrestrial space environment. Ionospheric changes are influenced by many factors, including solar radiation, geomagnetic activity, and meteorological influences [1,2]. Those factors can easily change the ionosphere abnormally within a short period. Such rapid variations can give rise to negative effects on several technologies, such as navigation and positioning error, radio communication failure, spacecraft damage, ground power transmission, and oil transmission system damage [3]. Since the 1960s, the ionosphere total electron content (TEC), as one of the important characteristic parameters of the ionosphere, has been extensively used in ionospheric research [4]. Analysis of temporal and spatial variations of the TEC can not only reduce the impact of the ionospheric delay but also further explore the coupling mechanism of the solar wind, magnetosphere, thermosphere, and ionosphere to make more rational use of the near-Earth

space environment [5,6]. At the same time, a large number of studies pointed out that the TEC has obvious diurnal, semiannual, and annual variations in time, and the changes in solar ionizing radiation, geomagnetic activity, and geographical location have significant impacts on the range of the TEC variation [7–10]. Due to geographical differences, there will also be local anomalies, such as the equatorial ionization anomaly (EIA) [11], Weddell Sea Anomaly (WSA) [12–14], and winter anomaly [7,8]. Therefore, studies and analyses are conducted on the TEC in the SAA under different solar activity conditions.

In recent years, the EOF method extracted the spatial and temporal variation characteristics of data well and was widely used in ionospheric research and empirical modeling. For example, Mao et al. [15] constructed a climatology model of the TEC over China based on EOF analysis using global positioning system (GPS) data. They found that the EOF model had better assimilation nowcasting capability when the EOF model and the International Reference Ionosphere (IRI) model were used in the three-dimensional variational (3DVAR) data assimilation experiment.

The EOF method of building a global ionospheric TEC model based on the global ionosphere maps (GIMs) was used, and it was confirmed that the model can reflect the temporal and spatial distribution characteristics of the TEC well when compared with the observed data [9]. Chen et al. [16] adopted the EOF method to analyze the characteristics of the GPS TEC and established the empirical TEC model of North America. They found that the EOF analysis could reflect the temporal–spatial characteristics of the TEC data well and was found to be an effective method for building empirical models and data analysis. Jamjareegulgarn et al. [17] utilized the TEC observation data of GPS stations in Nepal to conduct EOF analysis to research the annual and monthly changes of the TEC in Nepal. Then, they established an EOF model by using global geomagnetic activity works that can accurately depict TEC changes.

The South Atlantic Anomaly (SAA) is a strong geomagnetic field anomaly in the southern part of South America and the South Atlantic Ocean. Its magnetic field is 30% to 50% weaker than elsewhere at the same latitudes, causing more radiation from outer space to penetrate the atmosphere closer to the Earth's surface [18,19]. Communications of satellites, aircraft, and spacecraft in the region are more susceptible to interference [18], and the enhanced radiation poses a health risk to astronauts [20]. Although there are satellite missions that cover the SAA area, such as the Global-scale Observation of Limb and Disk (GOLD) [21], few studies have been carried out [22]. Zeng et al. [23] found that numerical simulations of the Thermosphere-Ionosphere-Electrodynamics General Circulation Model (TIEGCM) model fit well with ionospheric radio occultation (IRO) observations, with both reflecting a similar annual asymmetry of $NmF_2$, as well as similar semidiurnal and longitudinal variations. By comparing the performance of the TEC during solar cycle 24 using GPS, the IRI model, and the TIEGCM V2.0 model, Rao et al. [24] found that the observed TEC was almost identical with the simulated TEC, depicting similar trends in the solar cycle, semiannual, and seasonal variations. Therefore, it is interesting and novel to use the TEC data in the SAA obtained from TIEGCM simulations for spatial and temporal variation analysis.

In this study, the ionospheric TEC data obtained from TIEGCM simulations in 2002 and 2008 were processed using the EOF analysis method to discuss the spatial and temporal variation patterns of the TEC in the solar maximum and solar minimum over the SAA, which could provide further insight into the spatial and temporal variation characteristics of the TEC in the SAA. At the same time, this work fits in with the subsequent AUKO-1 satellite project in China and also lays the foundation for the comparison of the TEC predicted by the TIEGCM model with the observations of the TEC that are to be provided by the successful launch of the AUKO-1 satellite in the future.

Section 2 introduces the TIEGCM and EOF methodology. Section 3 gives the main results. In Section 4, we give the analysis results of the EOF and the differences between the solar maximum and solar minimum. The summary is given in Section 5.

## 2. Model and Method

### 2.1. TIEGCM Model

The Thermosphere-Ionosphere-Electrodynamics General Circulation Model (TIEG CM) [25–28] developed at the National Center for Atmospheric Research (NCAR) High-Altitude Observatory is a first-principle, time-dependent, three-dimensional model that self-consistently solves the energy, momentum, and continuity equations of the coupled thermosphere/ionosphere. It can self-consistently simulate many major parameters in the thermosphere and ionosphere, such as the neutral temperature, wind, and TEC. The primary external forcing of the TIEGCM consists of solar UV radiation expressed by the F10.7 index and geomagnetic forcing driven by the Kp index in the Heelis mode [29], where solar UV radiation is the main energy input. However, the geomagnetic energy input is comparable to or even larger than the solar radiation input during magnetic storms [27,30,31]. The lower boundary of the model is driven by the monthly climatology of the tide using the Global Scale Wave Model (SGWM) [32].

In this work, we used TIEGCM V2.0, which has a horizontal resolution of $2.5° \times 2.5°$ in geographic latitude and longitude, 57 pressure surfaces from ~97 km to ~500 km, with a vertical resolution of $\frac{1}{4}$ scale height [28]. We conducted the TIEGCM simulations in 2002 and 2008 with the model output cadence of 1 h. For the current study, we selected the TEC data in the SAA ($10°$ N to $60°$ S, $20°$ E to $100°$ W) to conduct a comparative analysis of the spatial and temporal variations in the TEC during the solar maximum and solar minimum.

### 2.2. EOF Analysis Method

Empirical orthogonal function (EOF) analysis, also known as principal component analysis (PCA), was originally proposed by the statistician Pearson [33] as a method to analyze the structural properties of matrix data and extract the main features of the data, which can decompose the original data into a set of uncorrelated basis functions and weight coefficients via an orthogonal transformation, which serves to describe most of the features of the original data using a small amount of data [34–38].

In this study, the TEC grid data (28 latitudes $\times$ 49 longitudes with a 1 h resolution) obtained from the TIEGCM simulations were first processed into spatially and temporally arranged data (1372 grid points $\times$ 8736 time points, where 8736 is the number of simulated values in the previous 364 days in 2002 or 2008), and the data were preprocessed [34], i.e., the mean value of each row of the matrix, which represents the time series of the hourly TEC at a specific coordinate, was removed, ensuring that each row had a zero mean. The processed data was noted as TEC($\theta$, $\varphi$, t), where ($\theta$, $\varphi$) represents the latitude and longitude position and t represents the time (the time interval was 1 h). The EOF analysis involved decomposing the TEC varying with time and space into a time function $T_i$ and space function $EOF_i$, where the EOFs were uncorrelated over space, as in the following equation:

$$\mathrm{TEC}(\theta, \varphi, t) = \sum_{i=1}^{n} \mathrm{EOF}_i(\theta, \varphi) T_i(t) \tag{1}$$

In Equation (1), $EOF_i(\theta, \varphi)$ is the ith order EOF basis function, which represents the spatial variation characteristics of the TEC; $T_i(t)$ is the time coefficient corresponding to the ith order EOF basis function, which describes how the spatial variability varies in time; n is the number of EOF level terms.

$$T_i(t) = \sum_{j=1}^{m} E_{ij} A_{ij} \tag{2}$$

In Equation (2), $T_i(t)$ is the ith order time coefficient, and the time coefficients are arranged by 24 h $\times$ 364 d (24 represents hours, 364 represents days) and then the EOF analysis was performed separately to obtain the basis function $E_{ij}$ and the coefficient $A_{ij}$. The function $E_{ij}$ denotes the diurnal variation in the time coefficient $T_i$, $A_{ij}$ denotes the annual variation in the time coefficient $T_i$, and m is the number of EOF level terms; only the first-order coefficient $A_{i1}$ was selected for analysis and discussion in this study.

The proportion of each eigenvalue $\lambda_i/\lambda_j$ in the sum of total eigenvalues is the variance contribution rate $C_i/C_j$, which reflects the proportion of a spatial typical field representing the total features, where the larger the variance contribution rate, the greater the weight of the spatial typical field. The formulas for calculating the variance contribution rate are as follows:

$$C_i = \frac{\lambda_i}{\sum_{k=1}^{n} \lambda_k} \tag{3}$$

$$C_j = \frac{\lambda_j}{\sum_{k=1}^{m} \lambda_k} \tag{4}$$

## 3. Results

The variance contribution rate formula (Equation (3)) was used to calculate the variance contribution rate of the first three EOFs obtained using the EOF decomposition of the TEC over the South Atlantic Ocean. The cumulative variance contribution rates of the first three EOFs in 2002 and 2008 were 94.80% and 93.86%, respectively, both of which were enough to reflect the overall data characteristics. The results were as follows: Table 1 shows the variance contribution rates of the first three EOFs obtained using the EOF decomposition of the TEC in 2002 and 2008. Equation (4) was used to obtain Table 2. Table 2 shows the variance contribution rates of the first mode obtained by the EOF decomposition of the first three time coefficients $T_i(t)$ (i = 1, 2, 3) in each of 2002 and 2008.

**Table 1.** Variance contribution of the first three $EOF_i$ (i = 1, 2, 3) of the TEC in each of 2002 and 2008.

| EOF Type | 2002 (%) | 2008 (%) |
|---|---|---|
| $EOF_1 \times T_1$ | 65.59% | 59.85% |
| $EOF_2 \times T_2$ | 25.23% | 29.98% |
| $EOF_3 \times T_3$ | 3.98% | 4.03% |

**Table 2.** Variance contribution of the first mode obtained by EOF decomposition of the first three time coefficients $T_i(t)$ (i = 1, 2, 3) in each of 2002 and 2008.

| EOF Type | 2002 (%) | 2008 (%) |
|---|---|---|
| $T_1: E_{11} \times A_{11}$ | 98.84% | 98.13% |
| $T_2: E_{21} \times A_{21}$ | 97.00% | 98.57% |
| $T_3: E_{31} \times A_{31}$ | 91.21% | 89.23% |

Figure 1 shows several indices that describe the solar radiation and geomagnetic conditions in 2002 (red lines) and 2008 (blue lines): (Figure 1a) F10.7 index, (Figure 1b) Ap index, and (Figure 1c) Dst index. The geomagnetic indices Ap and Dst and the solar EUV proxy F10.7 were averaged daily here. As can be seen from Figure 1a, the F10.7 index fluctuated between ~110 sfu and ~253 sfu in 2002. However, the change in the F10.7 index in 2008 was relatively stable, and the maximum value only reached 88.2 sfu. In 2002, there were two sharp decreases in the Dst index in April and October, while, correspondingly, there were two peaks in the Ap index. According to Figure 1b,c, it can be seen that the maximum value of Ap reached 77 and the minimum value of Dst reached $-120$ nT in 2002, while the fluctuation of the geomagnetic indices was significantly smaller in 2008 than that in 2002. Therefore, the solar EUV radiation was significantly higher in 2002 than that in 2008, and the geomagnetic activity was higher.

Figure 2a,b represent the TEC time-averaged plots for the SAA in 2002 and 2008, respectively; Figure 2c–h represent the spatial variation in the TEC for the SAA in 2002 and 2008, i.e., the first three EOFs: $EOF_i(\theta, \varphi)$ (i = 1, 2, 3). It should be pointed out that the first three EOFs only represent the spatial distribution features of the TEC variations in

the SAA; the values of the first three EOFs cannot be directly compared with the values of the TEC in the TEC time-averaged plots. Figure 3a–f show the time coefficients $T_i(t)$ (i = 1, 2, 3) corresponding to the first three EOFs in the SAA in 2002 and 2008, and the black curve superimposed on each figure is the annual variation curve $A_{i1}$ obtained via EOF decomposition of the time coefficients $T_i(t)$ (i = 1, 2, 3). The $A_{i1}$ annual variation curves corresponding to the first three time coefficients in 2002 and 2008 were correlated with F10.7, Dst, and Ap to obtain Table 3.

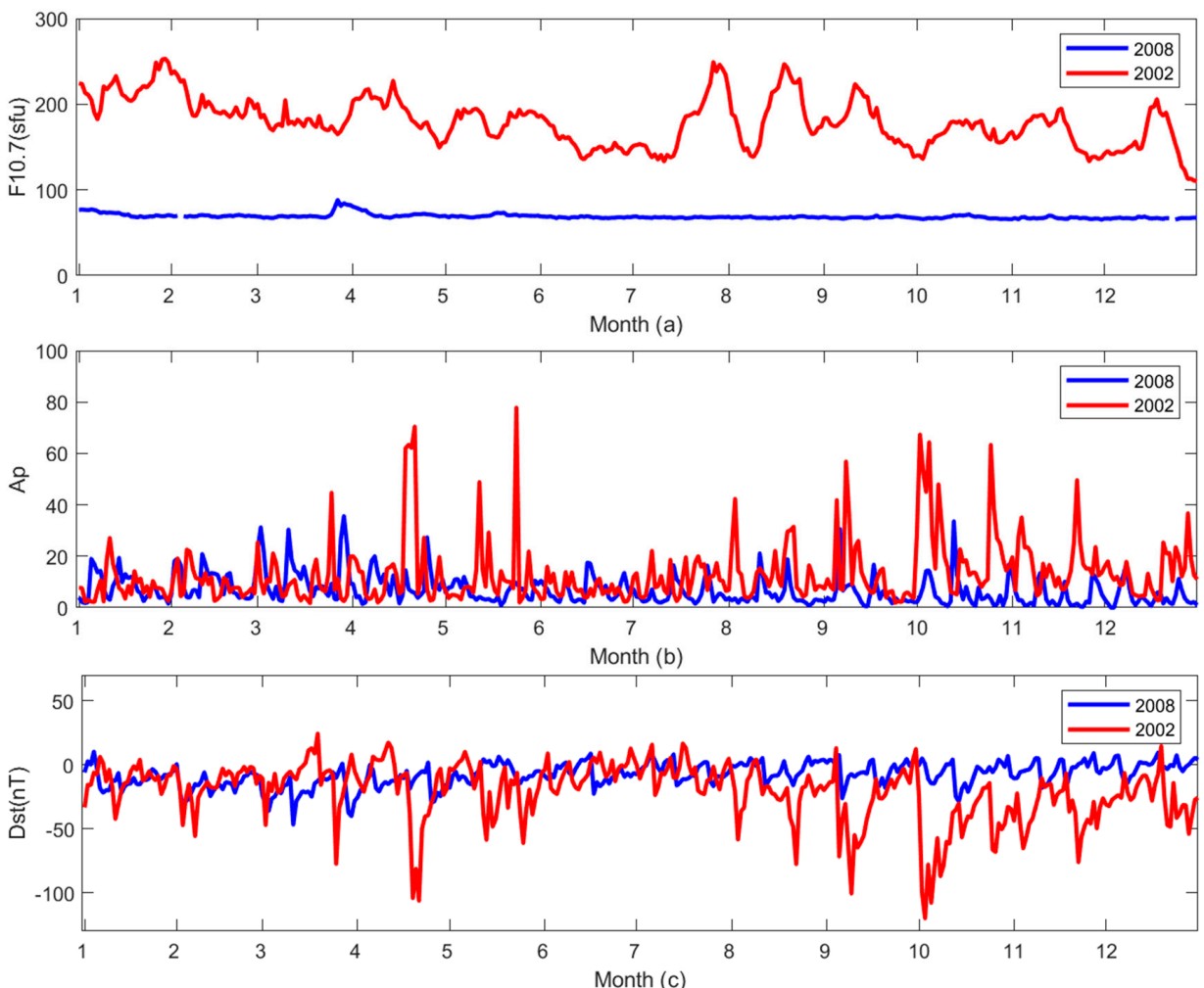

**Figure 1.** Solar radiation and geomagnetic conditions in 2002 (red lines) and 2008 (blue lines): (**a**) F10.7 index, (**b**) Ap index, and (**c**) Dst index.

**Table 3.** The correlation coefficients obtained from the correlation analysis of the $A_{i1}$ annual variation curve with Dst, F10.7, and Ap in 2002 and 2008.

|  |  | Dst | F10.7 | Ap |
|---|---|---|---|---|
|  | $T_1: A_{11}$ | −0.1401 | 0.7348 | 0.2116 |
| 2002 | $T_2: A_{21}$ | −0.1884 | 0.5109 | 0.0658 |
|  | $T_3: A_{31}$ | 0.148 | 0.2987 | 0.083 |
|  | $T_1: A_{11}$ | −0.7898 | 0.0336 | 0.6407 |
| 2008 | $T_2: A_{21}$ | −0.6706 | 0.0718 | 0.5307 |
|  | $T_3: A_{31}$ | −0.5247 | −0.0238 | 0.6021 |

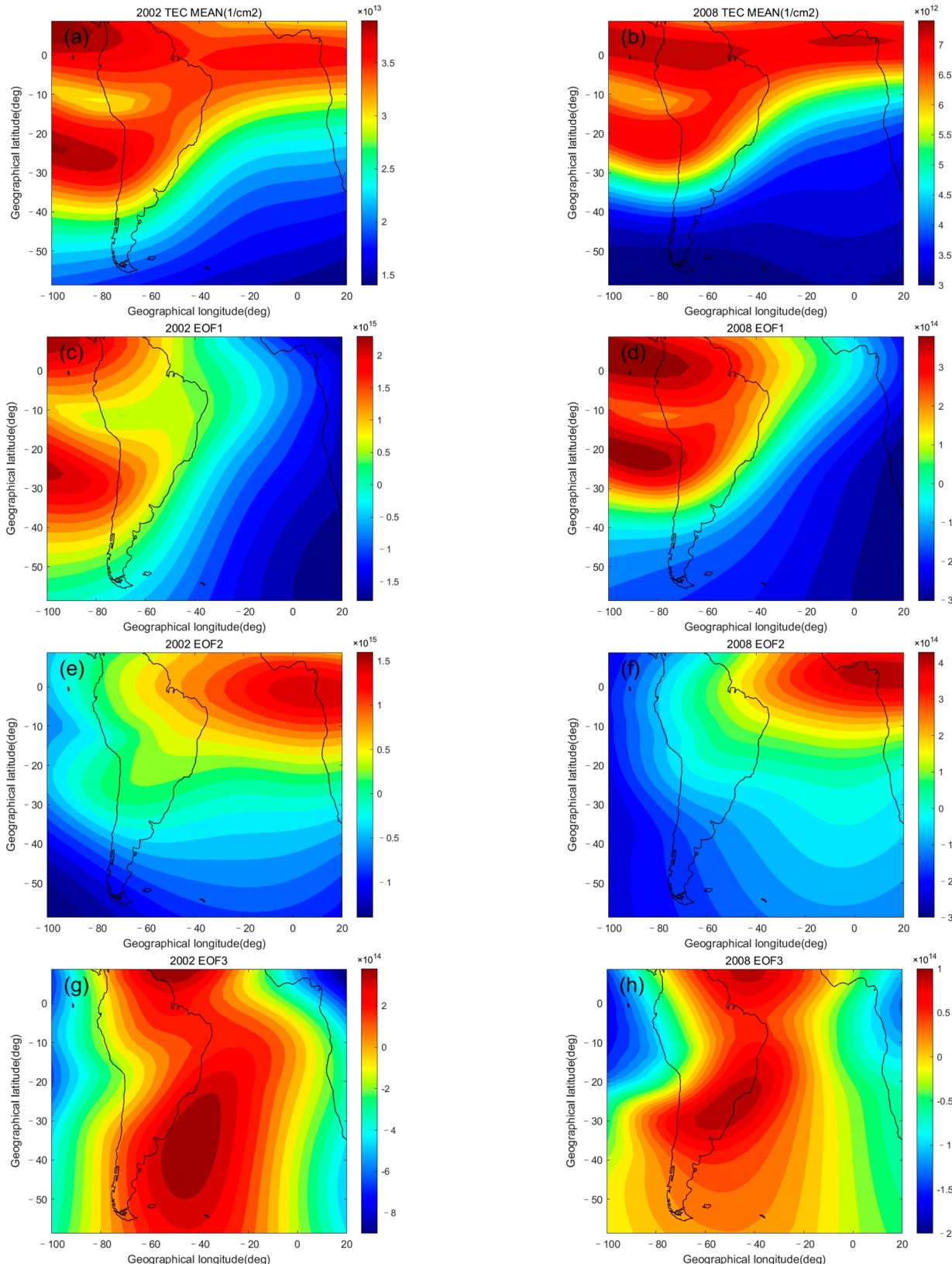

**Figure 2.** (**a**,**b**) TEC time-averaged maps ($1/cm^2$) derived from the TIEGCM simulations, and (**c**–**h**) the first three $EOF_i$ (i = 1, 2, 3) in the SAA in each of 2002 and 2008.

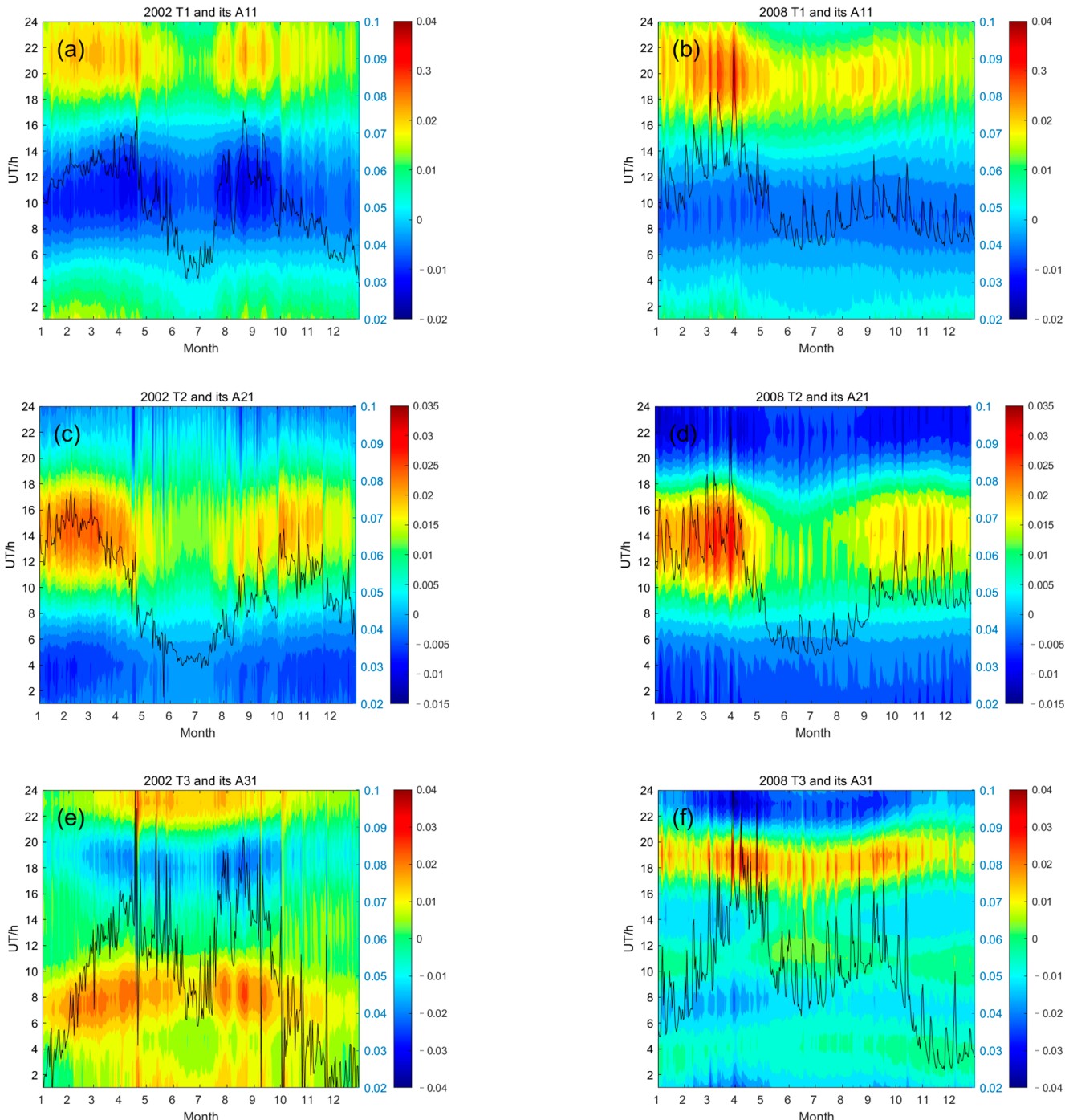

**Figure 3.** The diurnal and annual variations of time coefficients Ti(t) (i = 1, 2, 3) corresponding to the first three $EOF_i$ (i = 1, 2, 3) in the SAA in each of 2002 and 2008 are shown (color figure), and the annual variation curves $A_{i1}$ (black curve) obtained by performing an EOF on the corresponding time coefficients are superposed on the color figures.

## 4. Interpretation and Discussion

### 4.1. TEC Mean Field

Figure 2a,b display the time-averaged plots of the TEC in the SAA in 2002 and 2008. It can be seen that the maximum ionospheric TEC occurred near the geomagnetic equator, and the EIA occurred around 20° S~30° S and 80° W~100° W. We also found that the TEC in the SAA was influenced by solar EUV radiation, and the values of the ionospheric TEC was significantly higher in the solar maximum than that in the solar minimum (see color

bar). At the same latitude, the TEC on the west side of the SAA was higher than that on the east side in both 2002 and 2008, showing an east–west asymmetry. The east–west difference was larger during the solar maximum, which was also well reflected in $EOF_1$ (Figure 2c,d). At the same longitude, the eastern side of the SAA (40° W~20° E, Figure 2a,b) showed a monotonic decrease in the TEC with increasing latitude in 2002 and 2008, which was more evident in $EOF_2$ (Figure 2e,f).

Similar to our results, Kaufmann et al. [39] utilized the ATS-6 satellite observations data and found the east–west difference in the TEC in the SAA, and they concluded that the seasonal variation only accounted for 6% of the influence on the longitude variation in the TEC, and the change in magnetic inclination could not have a significant effect on the reduction in the TEC. The east–west asymmetry of the TEC has been a great controversy, with different researchers proposing different influencing factors. Massambani [40] suggested that the east–west asymmetry of the ionospheric TEC in the SAA is caused by the precipitation of low-energy electrons (E < 40 keV) during their longitudinal drift. Sojka et al. [41] considered that the longitudinal dependence of the daytime TEC enhancement at mid-latitudes during storms is related to thermospheric winds and electric fields. Meanwhile, Zhang et al. [42] proposed that the east–west asymmetry of the TEC is caused by a combination of variations in geomagnetic declination and zonal winds. These results may also apply to the longitudinal variation in the TEC in other geographic regions.

*4.2. First Mode*

The first mode EOF and its time coefficients are presented in Figures 2c,d and 3a,b, which represented 65.59% and 59.85% of the total variance in 2002 and 2008, respectively. $EOF_1$ (Figure 2c,d) reflects the most significant spatial feature of the TEC in the SAA. Similar to the time-averaged plots, there was a significant zonal variation near the EIA. The TEC values were significantly higher in the solar maximum than that in the solar minimum, and the maximum values of the ionospheric TEC in 2002 and 2008 occurred in the peak regions of the EIA (60° W~100° W, 10° N~0°, Figure 2c; 60° W~100° W, 20° S~30° S, Figure 2d). Therefore, under the influence of the solar EUV radiation, the peak values and extent of the EIA were larger in the solar maximum than that in the solar minimum, which was consistent with the results of Liu and Chen [43]. The equatorial ionization anomaly (EIA), also known as the Appleton anomaly, is mainly explained by the "fountain effect" mechanism, in which the E-layer atmospheric generator generates an eastward polarization field during the daytime under the action of the neutral wind, and the eastward polarization field is transmitted upward along the magnetic lines to the F-layer, causing an E × B force that drives the plasma upward at the geomagnetic equator. Then, the plasma diffuses downward along the magnetic lines and gravity precipitates to the north and south sides of the equator to form a bimodal structure [11].

It should be noted that $T_i$ and $A_{11}$ could not directly represent the time distribution of the TEC in 2002 and 2008, but could only represent the time distribution characteristics of the modes corresponding to the TEC changes. Therefore, $EOF_i(\theta, \varphi) \times T_i(t)$ should be discussed in conjunction with $EOF_i(\theta, \varphi)$ and $T_i(t)$ when analyzing the variation in the TEC. For example, Figure 2c,d showed that the east-side TEC was higher than the west-side TEC when $T_1(t)$ (Figure 3a,b) was negative in the morning, and the west-side TEC was higher than the east-side TEC when $T_1(t)$ was positive in the afternoon, showing the east–west asymmetry of the TEC in the SAA in 2002 and 2008.

As shown in Figure 3a,b, the time variations of $EOF_1$ showed the diurnal variation, which implied that the TEC increased after sunset in the solar maximum and solar minimum. Moreover, compared with the solar minimum, the amplitude of the TEC was more variable and the bimodal structure was more obvious in the solar maximum (combining $EOF_1(\theta, \varphi) \times T_1(t)$). These characteristics were consistent with the results of Zhao et al. [44] and Zhao et al. [8]. In terms of the annual variation, in 2002, the maximum values of $T_1$ (Figure 3a) mainly occurred in April and August, and the minimum values mainly occurred in June and December; in 2008, the maximum values of $T_1$ (Figure 3b) mainly

occurred in March and September, and the minimum values occurred in June~July and November~December. However, the spring–autumn asymmetry of the TEC only appeared in the solar minimum. It can also be seen that the TEC was obviously dependent on the seasons. The TEC over the South Atlantic had an obvious longitudinal enhancement in the spring and autumn in the solar maximum and solar minimum. These characteristics were consistent with the results of Scherliess et al. [45].

The $A_{11}$ annual variation coefficients corresponding to $T_1$ in 2002 and 2008 represented 98.84% and 98.13% of the total variance in $T_1$, respectively. The correlation showed that the annual TEC variation in the solar maximum was mainly influenced by the solar radiation F10.7, and the correlation coefficient with F10.7 was 0.7348. It was less influenced by geomagnetic activity, and the correlation coefficients with Dst and Ap were −0.1401 and 0.2116, respectively; the annual TEC variation in the solar minimum was mainly influenced by geomagnetic activity, and the correlation coefficients with Dst and Ap were −0.7898 and 0.6407, respectively, while there was almost no correlation with F10.7 (r = 0.0336). In general, the ionosphere TEC was mainly influenced by solar EUV radiation, as shown in the results of the study in the solar maximum of 2002. However, the correlation between $A_{11}$ and F10.7 was very small in the solar minimum, which was consistent with Wu's finding [46,47]. This phenomenon may have been due to the F10.7 index being relatively stable and having a small variation during the solar minimum. Another reason may be that $A_{11}$ only represents the annual variation characteristics of the TEC. Therefore, when calculating the correlation between $A_{11}$ and F10.7, the influence of solar radiation on the diurnal variation in the TEC was ignored, resulting in a smaller correlation coefficient.

*4.3. Second Mode*

Figure 2e,f show $EOF_2$, which accounted for 25.23% and 29.98% of the total variance in 2002 and 2008. The maximum values of the ionospheric TEC in 2002 and 2008 mainly occurred near the equator on the eastern side of the SAA (20° W~20° E, Figure 2e,f). Due to the influence of the solar EUV radiation intensity, the solar radiation intensity decreased with the increase in geographical latitude, which led to the decrease in TEC, showing the zonal variation in the TEC. The results indicated that intensive solar radiation had an important influence on the production of a large number of electrons in the middle and low latitudes, which was consistent with observations [10]. In addition, the values of the TEC and the active range of the zonal variation were larger in the solar maximum than that in the solar minimum. Finally, combined with $EOF_2(\theta, \varphi) \times T_2(t)$, it was found that the nighttime electron density in the southern summer (December~February) had an obvious enhancement phenomenon at mid-latitudes, especially near the Weddell Sea (60° S~70° S, 10° W~60° W, Figure 2e,f), namely, the WSA, also known as the mid-latitude summer nighttime anomaly (MSNA). This was consistent with the observations [12,13].

Figure 3c,d give the evolution of $EOF_2$'s time coefficients during 2002 and 2008. Diurnal variation is one of its notable variations, where it can be seen that the TEC in the SAA was affected by solar EUV radiation in 2002 and 2008, with a high TEC in the daytime and a low TEC at night. Except for the short-term perturbations, semiannual variation was most noteworthy. In 2002, the maximum values of $T_2$ (Figure 3c) appeared in February and October, and the minimum values appeared in June and July; in 2008, the maximum values of $T_2$ (Figure 3d) appeared in March and October~November and the minimum values appeared in June and July. The TEC exhibited a semiannual variation that was stronger in spring and autumn and weaker in summer and winter. Similar to the first mode (Figure 3a,b), the spring–autumn asymmetry of the TEC was more pronounced in the solar minimum. In addition, in the $EOF_2(\theta, \varphi) \times T_2(t)$ combination, it can be seen that the semiannual variation amplitude of the TEC was significantly greater in the solar maximum than that in the solar minimum, and the amplitude at the low latitude was larger than that at the middle latitude. These characteristics were consistent with the research results of Zhao et al. [44] and Ma et al. [48].

Although the annual variation in the TEC and $NmF_2$ is controlled by F10.7, the iono­spheric parameters at different altitudes are modulated differently by the solar activity, and the modulation amplitudes of the TEC and $NmF_2$ are obviously different [49]. However, Zhao et al. [8] and Liu et al. [50] found that the TEC has a similar diurnal variation, semian­nual variation, and annual variation to $NmF_2$, and the analysis results of the semiannual variation in the TEC and $NmF_2$ are roughly the same. Therefore, the possible mechanism of the semiannual variation in $NmF_2$ was used here to explain the semiannual variation in the TEC. Millward et al. [51] and Rishbeth et al. [52] proposed a mechanism that the combined [$O/N_2$], solar zenith angle, and atmospheric circulation changes at high lati­tudes, which may well explain the semiannual variation in $NmF_2$ in the far-pole regions. Meanwhile, the semiannual variation in $NmF_2$ has a regional characteristic, which can be interpreted differently with geomagnetic and geographic latitude and longitude changes. The semiannual variation is very weak in the near-pole regions and is mainly influenced by the variation in the [$O/N_2$]; the semiannual variation is strong in the far-polar regions, which usually peaks in March/April, and is affected not only by the semiannual variation in the [$O/N_2$] but also by the variation in the solar zenith angle. In addition, Ma et al. [48] suggested that the semiannual variation in $NmF_2$ is closely correlated with solar activity, and also suggested that the semiannual variation in $NmF_2$ in low latitudes is due to the presence of the equatorial electrojet with semiannual variation driven by the diurnal tide in the ionosphere, which then causes the semiannual variation in $NmF_2$ through the fountain effect of the ionosphere.

The $A_{21}$ coefficients corresponding to $T_2$ in 2002 and 2008 represented 97.00% and 98.57% of the total variance in $T_2$, respectively. The characteristics shown in terms of the correlation were similar to those of $T_1$, the annual variation in the solar maximum was mainly influenced by the solar radiation, and the correlation coefficient with F10.7 was 0.5109. $A_{21}$ was weakly influenced by geomagnetic activity, and the correlation coefficients with Dst and Ap were $-0.1884$ and $0.0658$, respectively; the annual variation in the solar minimum was mainly influenced by the geomagnetic activity; the correlation coefficients with Dst and Ap were $-0.6706$ and $0.5307$, respectively; and the correlation with solar radiation F10.7 was very poor (r = 0.0718).

*4.4. Third Mode*

$EOF_3$, which represented 3.98% and 4.03% of the total variance in 2002 and 2008, respectively, is shown in Figure 2g,h. Because the EOF decomposition method could not completely decompose the original data into a series of orthogonal basis functions, there was some data redundancy in the higher-order modes, resulting in the EIA of the TEC in both $EOF_1$ and $EOF_3$ in 2002 and 2008 (Figure 2c,d,g,h). In 2002, the peak of the EIA in the SAA mainly occurred in the range of 60° W~30° W and 20° S~50° S (Figure 2g). In 2008, the peak of the EIA in the SAA mainly occurred in the range of 65° W~40° W and 20° S~35° S (Figure 2h), indicating that the abnormal region was larger in the solar maximum than that in the solar minimum. It also showed that the intensity of the EIA was significantly greater in the solar maximum than that in the solar minimum due to the influence of the solar activity intensity.

Figure 3e,f give the evolution of $EOF_3$'s time coefficients during 2002 and 2008. The seasonal dependence of the TEC was reflected in the solar maximum and the solar mini­mum. In 2002, the maximum values of $T_3$ (Figure 3e) occurred in April and August, and the minimum values appeared in June~July and December. In 2008, the maximum values of $T_3$ (Figure 3f) occurred in April and September, while the minimum values appeared in June~July and December. Similar to the previous results, the TEC showed the spring–autumn asymmetry in the solar minimum. In addition, the winter anomaly is shown in Figure 3e,f, i.e., the ionospheric TEC was larger in winter (July) than in summer (January). Liu et al. [53] also found this feature in the TEC data obtained from GPS observations, and the phenomenon became more obvious with the increase in solar activity. Zhao et al. [8] and Yu et al. [7] considered that the [$O/N_2$] variability is the main contributor to TEC

winter daytime anomalies and is also responsible for some of the semiannual and annual anomalies. Due to pressure differences, upwelling air occurring at low latitudes and in the summer hemisphere decrease the $[O/N_2]$, and thus, decrease the electron content, while upwelling air brings molecule-enriched air to the mid-latitudes of winter hemispheres. the downwelling air will increase the $[O/N_2]$, thereby increasing electron content in the winter hemisphere [54].

The $A_{31}$ coefficients corresponding to $T_3$ represented 91.21% and 89.23% of the total variance in $T_3$ in 2002 and 2008, respectively. Although $A_{31}$ had a very weak correlation with F10.7 in 2002 (r = 0.2987), it had an even lower correlation with Dst (r = 0.148) and Ap (r = 0.083), thus indicating that the annual variation in the solar maximum was mainly affected by the solar radiation. In 2008, $A_{31}$ was moderately correlated with Dst (r = −0.5247) and Ap (r = 0.6021) but hardly correlated with F10.7 (r = −0.0238), indicating that the annual variation in the solar minimum was mainly affected by the geomagnetic activity.

## 5. Summary

This study focused on using the TEC data of the SAA in 2002 and 2008 obtained from the TIEGCM simulations to obtain the TEC time-averaged maps and the first three EOFs compared and analyzed the spatial and temporal variation in the TEC in the SAA in the solar maximum and solar minimum. In addition, a correlation analysis was performed using the annual coefficients to identify the possible drivers of the TEC changes. The results of the analysis showed the following:

(1) Similarities: Although the change in the intensity of the solar activity had a strong influence on the magnitude of TEC, it did not change the spatial and temporal characteristics of the TEC much. Whether it was in the solar maximum or solar minimum, spatially, $EOF_1$ and $EOF_3$ reflected the EIA influenced by plasma drift; in addition, $EOF_1$ also reflected the east–west asymmetry, which may have been related to the variation in the magnetic declination angle, and the zonal wind; $EOF_2$ mainly reflected the zonal variation in the TEC, which was highly correlated with the solar EUV radiation. Temporally, $T_1$ to $T_3$ mainly reflected the semiannual variation in the TEC, which could not be simply explained by the variation in the $[O/N_2]$, but the variation in the solar zenith angle and atmospheric circulation also provided significant contributions; $T_3$ reflected the winter anomaly influenced by the $[O/N_2]$. It also indicates that these distribution features were regular features of the TEC distribution over the SAA.

(2) Differences: The values and amplitude variation in the TEC, which were strongly modulated by the solar activity, were significantly greater in the solar maximum than that in the solar minimum. Under the influence of the solar EUV radiation intensity, the active range of the zonal variation and the EIA in the SAA also expanded with the increase in the solar activity intensity. The spring–autumn asymmetry was observed in the ionospheric TEC during the solar minimum. Furthermore, it can be seen from the correlation that the TEC was mainly influenced by the solar radiation (F10.7) in the solar maximum (r = 0.7348), while the TEC was mainly influenced by the geomagnetic activity (Dst) in the solar minimum (r = −0.7898).

Finally, the relative contributions of the EOFs (and the drivers with which they are associated) may be confirmed by satellite and other observations that are to be made under similar conditions of solar activity after the launch of the AUKO-1 satellite. The discrepancies between the model and the observations will indicate the deficiencies of the model and/or possible biases in the observations, and thus, facilitate the subsequent improvement of the model.

**Author Contributions:** Conceptualization, J.Y. and Z.L.; methodology, J.Y., Z.L. and Y.W.; software, J.Y. and Y.W.; validation, Z.L. and Y.W.; formal analysis, J.Y., Y.W., J.S. and L.W.; investigation, J.Y. and Z.L.; resources, Z.L.; data curation, J.Y. and Z.L.; writing–original draft preparation, J.Y.; writing—review & editing, J.Y., Z.L., Y.W., J.S., L.W., J.L., H.Z., X.X. and C.G.; visualization, J.Y.;

supervision, Z.L. and J.L.; project administration, Z.L., J.L. and X.X.; funding acquisition, Z.L. All authors have read and agreed to the published version of the manuscript.

**Funding:** This research was funded by the National Natural Science Foundation of China (grant numbers 42074183 and 42004132) and the Open Project Program of State Key Laboratory of Lunar and Planetary Sciences (Macau University of Science and Technology) (Macau FDCT grant no. SKL-LPS(MUST)-2021-2023).

**Data Availability Statement:** The F10.7, Ap and Dst indices are provide by the OMNI database (https://cdaweb.gsfc.nasa.gov/, accessed on 2 January 2023). The simulation data presented in this study are available on request from the corresponding author. The data are not publicly available due to privacy.

**Conflicts of Interest:** The authors declare no conflict of interest.

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
