# Peer review of "An Empirical Orthogonal Function Study of the Ionospheric TEC Predicted Using the TIEGCM Model over the South Atlantic Anomaly in 2002 and 2008"

_universe, doi:10.3390/universe9020102_

Round 1

Reviewer 1 Report

See attached zip-file containing the following:

1. Universe 2171159 Yu et al  General remarks.pdf

Author Response

Comments:

General Comments: The paper uses vertical TEC (vTEC) values over the South Atlantic (Magnetic)

Anomaly (SAA) as simulated by the TIEGCM model for a comparison of the spatial and temporal

characteristics of TEC during solar maximum (2002) and solar minimum (2008).

The paper mentions similar comparisons based on observations, but justifies the present work by the following two claims:

1) In the Introduction it is claimed that “the analysis of model data is very rare”. This claim is not

true. Analysis of data generated by ionospheric models is not rare. There are many papers that

analyse the TIEGCM model by comparing it to observations. See the list of references below. Only

one of these papers are mentioned.

Reply:

Thank you for your comments. I agree with you that analysis of data generated by ionospheric models is not rare. Therefore, this sentence has been removed in the revised article.

Comments:

2) In the Summary it is stated that “it is hoped that the results of this work can be compared with the observations provided by the successful launch of our future AUKO-1 satellite”. The planned comparison would indeed be a valuable addition to the knowledge base regarding the usefulness and accuracy of the TIEGCM model under different conditions of solar activity. Without such validation of the TIEGCM model over the region of interest, the analysis of the ionospheric TEC predicted by the model, is of limited value. The results presented in this paper pertains to historical dates in 2002 and 2008. Ionospheric models are useful for prediction of ionospheric conditions, i.e. to provide data at future timess when observations are not yet available (Jakowski et al., 2011). The use of a model to compare historical TEC characteristics while observations are available, makes no sense. It will not be possible to “compare the results of this work” i.e. “the variability of the TEC” in 2002 and 2008 with observations of the AUKO-1 satellite, since the satellite is yet to be launched. However, it may be shown that the relative contributions of the EOFs (and the drivers with which they are associated) may be confirmed by satellite and other observations to that are to be made under similar conditions of solar activity after the launch of the AUKO-1 satellite.

Reply:

Thank you for your comments. In this paper, due to the special weak magnetic field conditions in SAA region, we used TIEGCM to obtained the TEC in SAA region for comparative analysis of the spatial and temporal characteristics of solar maximum (2002) and solar minimum (2008), as well as research on the driving factors affecting the change of TEC under different solar activity conditions. Your comments gave me a lot of inspiration. I agree with you and accept your suggestions to revise the language expression of the last part in the revised edition of our paper. We hope to use the EOF method to extract the temporal and spatial characteristics of TEC in SAA region and analyze the correlation, and the obtained results can be verified and compared with the observation results of AUKO-1 satellite under similar solar activity conditions in the future. The comparison between the model data and the observation results of AUKO-1 satellite is not the comparison of TEC data in 2002 and 2008.

Comments:

No basis is provided for the validity of the TIEGCM model over the SAA nor is it compared to other ionospheric models that have been validated over the SAA. Before the TIEGCM model can be used in the SAA there should have been studies to demonstrate that the models adequately describe the spatial and temporal characteristics of the ionosphere in this region. The statement

in the Conclusions make it clear that the authors expect the ITEGMC model to have deficiencies over the region of interest. Hence it is not clear why simulated data would provide a better comparison of TEC in 2002 and 2008 over the SAA than observations.

Reply:

Thank you for your comments. I have benefited a lot. This paper focuses on using TIEGCM model to simulate TEC data for EOF analysis and comparison in the SAA in 2002 and 2008. TIEGCM has been widely used, and it has been pointed out in previous papers that the model fully describes the space-time characteristics of the ionosphere, but no application has been found in the SAA region. Therefore, We have added model comparison content in the introduction section to demonstrate the effectiveness of the TIEGCM model.

Comments:

The title “Comparisons of the variability of ionospheric TEC” implies that the paper presents new knowledge about the behaviour of the ionosphere over the SAA. This is misleading. It should be clearly stated that the paper presents an EOF evaluation of the TEC predicted by the TIEGCM model for the purpose of identifying the potential drivers of the variability of the TEC over the SAA, thus establishing a set of parameters that would be examined by TEC observations by the AUKO-1 satellite. Suggested titles are the following: “A Principal Component Analysis of the ionospheric TEC based on TIEGCM simulations over the South Atlantic Anomaly in 2002 and 2008” or “An Empirical Orthogonal Function study of the ionospheric TEC predicted by the TIEGCM model over the South Atlantic Anomaly in 2002 and 2008”.

Reply:

Thank you for your advice. The previous title did cause some misunderstanding. We will change the title to “An Empirical Orthogonal Function study of the ionospheric TEC predicted by the TIEGCM model over the South Atlantic Anomaly in 2002 and 2008”.

Comments:

The paucity of measured data over the SAA during the periods of interest may be a reasonable

justification for the use of the TIEGCM model. If this is the case, such paucity needs to demonstrated.

Reply:

Thank you for your comments. It has already been noted in the revised paper that although there are satellite missions covering the SAA region (GOLD, Eastes et al., 2020), little research has been carried out (Cai et al., 2022).

Comments:

The graphics are of good quality. The scales of figures that compare 2002 and 2008 results should be the same to facilitate comparison. In some cases, the figure captions could be expanded to improve the explanations of the graphics.

Reply:

Thank you for your comment. We have considered setting the color bar in the same range before. However, When the scales of figures that compare 2002 and 2008 results is the same, the spatial characteristics of 2008 will not be well expressed, so the same color bar was not set. Considering your suggestion, we set the same ratio for Figure 2 in the revised version of the paper, but Figure 1 remains unchanged due to the poor effect after modification. And the title of the image has been modified.

Comments:

The language needs significant editing.

Reply:

Thank you for your comments. We rechecked the article and made some improvements of English expression.

Comments:

References must be given in the MDPI style, and not in IEEE style. See https://mdpi-res.com/data/mdpi_references_guide_v5.pdf Detailed comments are provided as annotations on the*.pdf version of the manuscript.

Reply:

Thank you for your comments. We have reviewed the references and revised them.

Reviewer 2 Report

This paper employs TIEGCM to generate the TEC in 2002 and 2008. They process the simulated TEC with EOF to get a bunch of coefficients. Then they analyze these coefficients and correspond these coefficients to the real physical meaning. The paper needs major revision before it is suitable for publication

Major comments 1: as we all know, the TEC mostly share similar pattern with NmF2, especially when consider them a large time scale. The author shall at least provide some statement on the reason to choose TEC, not to choose nmF2

Major comments 2: there are already numerous observations of TEC and NmF2 by low-earth orbiting satellites. These results include globally, and not to mention the SAA area. The author shall at least compare their model results with the previous observations. For example, in some certain LT frame, compare the pattern in lat-lon with previous studies such as Tulsi Ram et al., 2009. If the model cannot even reproduce the observation climatology, what is the meaning of your study based on unreliable results

Major comments 3: for the EOF method used, the reader had better know what EOF method are used previously for TEC and what results they obtain, any disadvantage from this current study?? Or whether it is necessary to carry out EOF on the TEC in SAA area

Major comments 4: the references in the paper are chosen pretty randomly. There are a bunch of new papers recently, but the author chooses to ignore, and randomly cite many old, out-dated papers that some of them are not even related to their goal!

Line 40 remove ‘ionization

Line 41 change the ionosphere abnormally

Line 45 ionosphere total electron content (TEC)

Line 52 have significant impacts on

Line 55 equatorial ionization anomaly (EIA). Since the author mention EIA and WSA here, they shall provide the related references.

Line 56-85 here the author shall also acknowledge some recent launched satellite missions that can cover SAA area, such as the Global-scale Observation of Limb and Disk (GOLD, Eastes et al., 2020), which is the first satellite in geo-stationary orbit to carry out studies in thermosphere and ionosphere. There is a recent work done by Cai et al., 2022 with GOLD and WACCM-X, who found the EIA interhemispheric asymmetry movement in this SAA area and it is due to the asymmetry of neutral wind and EXB drifts

Eastes, R. W., McClintock, W. E., Burns, A. G., Anderson, D. N., Andersson, L., Aryal, S., et al. (2020). Initial observations by the GOLD mission. Journal of Geophysical Research: Space Physics, 125, e2020JA027823. https://doi.org/10.1029/2020JA027823

Cai, X., Qian, L., Wang, W., McInerney, J. M., Liu, H.-L., & Eastes, R. W. (2022). Hemispherically asymmetric evolution of nighttime ionospheric equatorial ionization anomaly in the American longitude sector. Journal of Geophysical Research: Space Physics, 127, e2022JA030706. https://doi.org/10.1029/2022JA030706

The author can state that although there are satellite missions that cover SAA area (GOLD, Eastes et al., 2020), few studies have been carried out (Cai et al., 2022)

Additionally, I do not think there are only these observation studies, the author shall also cite some previous studies that focus on the global distribution of TEC, and check the SAA area TEC pattern.

Furthermore, why the author want to study both solar max and solar min?? they shall also emphasize the necessities of the comparison of solar max and solar min.

Line 85 please give the full name of TIEGCM, and shall provide the institute where the TIEGCM belongs to (NCAR)

Line 107 here the author shall provide the settings of solar irradiance or at least cite related papers to clarify the settings of solar irradiance

 Additionally, the settings of ionization or auroral precipitation shall also be mentioned, please check the recent published papers that using TIEGCM 2.0

Line 111-114 here the author just needs to state that the lower boundary of the model is driven by the monthly climatology of tide and cite the

Line 128-129 EOF is widely used and getting some good results, so what are they?? Any examples??

Line 131-133 so the author just pick the area of SAA to process. Is there any difference for using the global model output with the same procedure??

 Results

The author shall first give the geomagnetic conditions in 2002 and 2008, together with solar irradiance. I recommend the author employ Kp and F10.7 to form the plots. Now the readers even do not know how the geomagnetic conditions differ in these 2 years!

Line 199 here this is not clear, what is meaning by near the equator??  Geomagnetic equator or geographic equator?? Near?? How much lat or lon??

Line 202 here the sentence shall be ended, and then start a new sentence from ‘At the same latitude,’

Line 203 the size of TEC?? What do you mean

Line 210 remove ‘as early as 1976’

Line 218 during the longitudinal drift?? This does not make sense, please re-write

Line 233-239 the statement about the EIA formation is not much related to your previous statement. You just mention the TEC value is larger in solar max than solar min, then you shall just explain why it is.

For the Huang and Cheng 1996, they focus in Asia area and it is too old from now. Actually there are many recent papers on the global climatology of TEC by LEO satellites, and the author shall consider use any of them to replace Huang and Cheng 1996, such as Liu and Chen

Liu, L., and Y. Chen (2009), Statistical analysis of solar activity variations of total electron content derived at Jet Propulsion Laboratory from GPS observations, J. Geophys. Res., 114, A10311, doi:10.1029/2009JA014533

Author Response

Major Comments 1:

This paper employs TIEGCM to generate the TEC in 2002 and 2008. They process the simulated TEC with EOF to get a bunch of coefficients. Then they analyze these coefficients and correspond these coefficients to the real physical meaning. The paper needs major revision before it is suitable for publication

as we all know, the TEC mostly share similar pattern with NmF2, especially when consider them a large time scale. The author shall at least provide some statement on the reason to choose TEC, not to choose nmF2

Reply:

Thanks for your comments on our paper. “Since the 1960s, ionosphere total electron content (TEC), as one of the important characteristic parameters of the ionosphere, has been extensively used in ionospheric research (Nelson, 1968). Analysis of temporal and spatial variations of TEC can not only reduce the impact of ionospheric delay (Jakowski et al., 2011) but also further explore the coupling mechanism of solar wind, magnetosphere, thermosphere, and ionosphere, so as to make more rational use of the near-Earth space environment (Mendillo, 2006). ” This is the reason why I choose to study TEC, which is reflected in the article. But we neglected to explain the reason of NmF2. Thank you for your reminding. We improved the 4.3 second model to express the difference between TEC and NmF2. For example, Nmf2 is the maximum electron concentration in the ionospheric F layer, and TEC is an integral parameter. Although the climate anomaly variation of TEC is similar to that of NmF2, and the annual variation is controlled by solar F10.7, the modulation amplitude of TEC and NmF2 is obviously different, indicating the difference of ionospheric parameters modulated by solar activities at different altitudes (Chen et al., 2012).

Chen, Y., Liu, L., Wan, W., & Ren, Z. (2012, March). Equinoctial asymmetry in solar activity variations of NmF2 and TEC. In Annales geophysicae (Vol. 30, No. 3, pp. 613-622). Copernicus GmbH.

Major Comments 2:

there are already numerous observations of TEC and NmF2 by low-earth orbiting satellites. These results include globally, and not to mention the SAA area. The author shall at least compare their model results with the previous observations. For example, in some certain LT frame, compare the pattern in lat-lon with previous studies such as Tulsi Ram et al., 2009. If the model cannot even reproduce the observation climatology, what is the meaning of your study based on unreliable results

Reply:

Thank you for your comments. At present, our objective in this paper is to study the characteristics of TIEGCM model and to analyze the characteristics of TEC in the SAA by using EOF method. Comparison of models and observations will be our goal in the future. The relative contributions of the EOFs (and the drivers with which they are associated) may be confirmed by satellite and other observations that are to be made under similar conditions of solar activity after the launch of the AUKO-1 satellite.

Major Comments 3:

for the EOF method used, the reader had better know what EOF method are used previously for TEC and what results they obtain, any disadvantage from this current study?? Or whether it is necessary to carry out EOF on the TEC in SAA area.

Reply:

Thank you very much for your comments. In the introduction part of the paper, we have added the introduction of the scholars' use of EOF method in TEC and the results obtained. Many papers have used EOF method to analyze the characteristics of TEC, mainly concentrated in parts of the northern hemisphere countries or global regions, such as China, South Korea, Nepal, North America, etc. The results show that the EOF method can reflect the temporal and spatial characteristics of the data. Due to the weak geomagnetic in the SAA region, it is also interesting to conduct EOF analysis of TEC in this region. The correlation analysis of TEC only analyzed the data of 2002 and 2008. It is not clear whether the same results will be produced by correlation analysis under long-term data. Long-term data (such as the 11-year solar cycle) can be used for correlation analysis in the future.

Major Comments 4:

the references in the paper are chosen pretty randomly. There are a bunch of new papers recently, but the author chooses to ignore, and randomly cite many old, out-dated papers that some of them are not even related to their goal!

Reply:

Thanks for your comments, we reviewed the article and made changes to the literature. We have deleted some references (Fürst et al., 2009; Casadio & Arino, 2011; Domingos et al., 2017; Abdu et al., 2005) and added new ones in the revised paper (Mao et al., 2008; Ercha et al., 2012; Chen et al., 2015; Jamjareegulgarn et al., 2020; Liu and Chen, 2009).

Comments:

Line 56-85 here the author shall also acknowledge some recent launched satellite missions that can cover SAA area, such as the Global-scale Observation of Limb and Disk (GOLD, Eastes et al., 2020), which is the first satellite in geo-stationary orbit to carry out studies in thermosphere and ionosphere. There is a recent work done by Cai et al., 2022 with GOLD and WACCM-X, who found the EIA interhemispheric asymmetry movement in this SAA area and it is due to the asymmetry of neutral wind and EXB drifts

The author can state that although there are satellite missions that cover SAA area (GOLD, Eastes et al., 2020), few studies have been carried out (Cai et al., 2022)

Additionally, I do not think there are only these observation studies, the author shall also cite some previous studies that focus on the global distribution of TEC, and check the SAA area TEC pattern.

Furthermore, why the author want to study both solar max and solar min?? they shall also emphasize the necessities of the comparison of solar max and solar min.

Reply:

Thank you for your comments. We have pointed out in the revised paper that there are satellite missions covering the SAA region, but few studies have been carried out. We have highlighted the need to study TEC under different solar activity conditions.

Comments:

Line 131-133 so the author just pick the area of SAA to process. Is there any difference for using the global model output with the same procedure??

Reply:

Thank you for your comments. there are many papers for EOF analysis of global TEC data, such as Ercha et al. (2012). Compared with regional TEC studies (Mao et al., 2008; Chen et al., 2015; Jamjareegulgarn et al., 2020), the EOF analysis of regional TEC can express the temporal and spatial characteristics of a place in more detail, and the temporal variation characteristics will also change with regional changes.

Comments:

The author shall first give the geomagnetic conditions in 2002 and 2008, together with solar irradiance. I recommend the author employ Kp and F10.7 to form the plots. Now the readers even do not know how the geomagnetic conditions differ in these 2 years!

Reply:

Thanks for your suggestion, we have added the introduction of geomagnetic index and F10.7 in the form of figure in the revised version.

Comments:

Line 40 remove ‘ionization

Line 41 change the ionosphere abnormally

Line 45 ionosphere total electron content (TEC)

Line 52 have significant impacts on

Line 55 equatorial ionization anomaly (EIA). Since the author mention EIA and WSA here, they shall provide the related references.

Line 85 please give the full name of TIEGCM, and shall provide the institute where the TIEGCM belongs to (NCAR)

Line 107 here the author shall provide the settings of solar irradiance or at least cite related papers to clarify the settings of solar irradiance

 Additionally, the settings of ionization or auroral precipitation shall also be mentioned, please check the recent published papers that using TIEGCM 2.0

Line 111-114 here the author just needs to state that the lower boundary of the model is driven by the monthly climatology of tide and cite the

Line 128-129 EOF is widely used and getting some good results, so what are they?? Any examples??

 Results

Line 199 here this is not clear, what is meaning by near the equator??  Geomagnetic equator or geographic equator?? Near?? How much lat or lon??

Line 202 here the sentence shall be ended, and then start a new sentence from ‘At the same latitude,

Line 203 the size of TEC?? What do you mean 

Line 210 remove ‘as early as 1976’

Line 218 during the longitudinal drift?? This does not make sense, please re-write

Line 233-239 the statement about the EIA formation is not much related to your previous statement. You just mention the TEC value is larger in solar max than solar min, then you shall just explain why it is.

For the Huang and Cheng 1996, they focus in Asia area and it is too old from now. Actually there are many recent papers on the global climatology of TEC by LEO satellites, and the author shall consider use any of them to replace Huang and Cheng 1996, such as Liu and Chen

Reply:

Thank you for your guidance and suggestions. Your patient guidance has greatly improved our paper. Through your comments, we have carefully reviewed the article  and revised the statements one by one.

Reviewer 3 Report

This work presents an interesting analysis based on TEC data obtained from TIEGCM in the region of the South Atlantic Anomaly considering a year of maximum solar activity level and a year of minimum. As such, I think it can be accepted for publication in this journal, but there are some results that are not clear which I comment below together with other observations.

Main comments:

1) I do not understand Table 2. Shouldn’t the variances add to 100%? Why they are all higher than 90%? I am sure a I am misunderstanding something here.

2) What does the annual coefficient mean, Aj, since we do not have inter-annual variation. I understand that you analyze only 2002 and 2008.

3) In the Discussion section:

The decrease to the west may be due to the magnetic equator at those longitudes is upward the geographic equator and you do not catch it in the region you are considering.

You mention here: “We also found that TEC in the SAA is influenced by solar EUV radiation, and the values of ionospheric TEC is significantly higher in solar maximum than that in solar minimum (see color bar); at the same latitude, the size of the TEC on the west side of the SAA is higher than that the east side in 2002 and 2008, showing an east-west asymmetry,  ...”

The EUV influence over TEC is logical due to higher ionization production, and the asymmetry towards the west is, again, due to the TEC spatial variation is shown in geographical coordinates, while it follows geomagnetic coordinates. In particular, at the upper region of the area you are considering since you include there the EIA. So for me, it is obvious. I do not agree with your sentence: “so the zonal variation is not particularly obvious.”

4) Regarding the second mode, I understand the TEC decrease with increasing latitude (as is also seen in the first mode), but I do not see the reason for the increase towards the East. Why is this?

5) In Table 3, what are the correlation coefficients if instead of the Aj’s you consider directly your TEC time series? For example, TEC time series obtained by averaging the whole area and the 24 hours for each day. Are they really different than those when you consider Aj? I understand the Aj time series is a daily mean series, with 365 points. Am I right?

Minor comments:

1) In line 39 you mention that the ionosphere “... has a protective role for the Earth”. Could you mention one, for example? I now it absorbs great part of EUV solar radiation, but I have never thought of it as a protective role. Or maybe you can give a reference to this. What can have negative effects are space weather conditions, which you mention in the following sentence.

2) In figure 1, it would be nice to add some contour lines of the Earth’s magnetic field intensity to delineate the South Atlantic Anomaly, for example the 2800 nT line. And if the map is in geographic coordinates, you could also add the map in black line in order to better detect the world region.

 3) Line 63: “KAUFMANN et al.(1976)” should be “Kaufmann et al. (1976)”

 4) Line 84: “Therefore, it is necessary to use the TEC data in the SAA obtained from TIEGCM simulations for spatial and temporal variation analysis.” I do not think that “necessary” is the correct word here. Maybe it could be “interesting and novel”. But I leave this to the authors to decide.

 5) Line 210: “KAUFMANN et al. (KAUFMANN et al., 1976)” should be “Kaufmann et al. (1976)”

 6) Line 328: I think that “it weakly influenced by ...” should be “It is weakly influenced by ...”

7) The reference list is arranged according to number, which are not used in the text. You should choose to use numbers (and in that case the references in the manuscript should be corrected according to this, or otherwise delete the numbers of each reference in the reference list and arrange them alphabetically. In fact, you should check the journals requirements for this.

Author Response

Comments:

This work presents an interesting analysis based on TEC data obtained from TIEGCM in the region of the South Atlantic Anomaly considering a year of maximum solar activity level and a year of minimum. As such, I think it can be accepted for publication in this journal, but there are some results that are not clear which I comment below together with other observations.

I do not understand Table 2. Shouldn’t the variances add to 100%? Why they are all higher than 90%? I am sure a I am misunderstanding something here.

Reply:

Thanks for your comments on our paper. In this paper, the data in Table 2 is the variance contribution rate corresponding to the first mode obtained by EOF analysis on the time coefficients T1, T2 and T3 obtained in Equation 1. For example, 98.84% is the variance contribution rate of the first mode obtained by EOF analysis on the T1  in 2002. 97.00% refers to the variance contribution rate of the first mode obtained by EOF analysis on the T2 in 2002, and so on. It has the same meaning as Table 1, but Table 1 is the contribution rate obtained by EOF on TEC. The variance contribution rate of the first mode basically reaches 90.00%, so only the first mode is selected for correlation analysis.

Comments:

What does the annual coefficient mean, Aj, since we do not have inter-annual variation. I understand that you analyze only 2002 and 2008.

In Table 3, what are the correlation coefficients if instead of the Aj’s you consider directly your TEC time series? For example, TEC time series obtained by averaging the whole area and the 24 hours for each day. Are they really different than those when you consider Aj? I understand the Aj time series is a daily mean series, with 365 points. Am I right?

Reply:

Thanks for your comments on our paper. Aj in this paper is obtained by EOF analysis of the time coefficient Ti, while Aj represents the time coefficient corresponding to the j-th Ej basis function, describing the change of diurnal variation in time. You are right that it is similar to the daily mean sequence. The annual variation refers to the change in 2002 and 2008.

In Table 3, if the time series of TEC is directly considered, the correlation coefficient will be too small. For example, TEC time series obtained by averaging the whole area and the 24 hours for each day, the correlation coefficient will be smaller. In 2002, the correlation coefficients between TEC and F10.7, DST and Ap were 0.229, 0.0399 and -0.0557, respectively.

Comments:

In the Discussion section:

The decrease to the west may be due to the magnetic equator at those longitudes is upward the geographic equator and you do not catch it in the region you are considering.

You mention here: “We also found that TEC in the SAA is influenced by solar EUV radiation, and the values of ionospheric TEC is significantly higher in solar maximum than that in solar minimum (see color bar); at the same latitude, the size of the TEC on the west side of the SAA is higher than that the east side in 2002 and 2008, showing an east-west asymmetry,  ...”

The EUV influence over TEC is logical due to higher ionization production, and the asymmetry towards the west is, again, due to the TEC spatial variation is shown in geographical coordinates, while it follows geomagnetic coordinates. In particular, at the upper region of the area you are considering since you include there the EIA. So for me, it is obvious. I do not agree with your sentence: “so the zonal variation is not particularly obvious.”

Reply:

Thank you for your comments. EIA, zonal variation, and east-west asymmetry are shown in Figure 1a,b. In the east side of the SAA region, TEC decreased with increasing latitude, namely, zonal variation. In the western part of the SAA region, the lower latitudes show equatorial anomalies, and compared with the eastern part, the zonal variation is weakened by the equatorial anomalies, but they are still evident. I agree with you and accept your suggestion that the wording of "so the zonal variation is not particularly obvious." is inappropriate and has been deleted in the revised version of the article.

Comments:

Regarding the second mode, I understand the TEC decrease with increasing latitude (as is also seen in the first mode), but I do not see the reason for the increase towards the East. Why is this?

Reply:

Thank you for your comments. We agree with your comments that zonal variation can also be seen in some areas of EOF1, which we have modified in the EOF1 results section of the revised version. It is normal for EOF2 to increase to the east. In Yao's article (2015), EOF1 also has a similar performance, but the physical mechanism has not been understood, so it is not explained.

Yao Xin;Zhao Biqiang;Liu Libo;Wan Weixing. Comparison of Ionospheric Total Electron Content over North America and East Asia with EOF Analysis (in Chinese). Chin. J. Space Sci., 2015, 35(5):556-565, dio:10.11728/cjss2015.05.556

Comments:

In line 39 you mention that the ionosphere “... has a protective role for the Earth”. Could you mention one, for example? I now it absorbs great part of EUV solar radiation, but I have never thought of it as a protective role. Or maybe you can give a reference to this. What can have negative effects are space weather conditions, which you mention in the following sentence.

Reply:

Thank you for your comments that let me find the problem and modify the content of the article.

Comments:

In figure 1, it would be nice to add some contour lines of the Earth’s magnetic field intensity to delineate the South Atlantic Anomaly, for example the 2800 nT line. And if the map is in geographic coordinates, you could also add the map in black line in order to better detect the world region.

Reply:

Thank you for your comments. We accept your suggestion and have added the map outline with a black line to Figure 1 in the modified version.

Comments:

Line 63: “KAUFMANN et al.(1976)” should be “Kaufmann et al. (1976)”

Reply:

Thanks for your comments. The sentence was amended in the revised paper.

Comments:

Line 84: “Therefore, it is necessary to use the TEC data in the SAA obtained from TIEGCM simulations for spatial and temporal variation analysis.” I do not think that “necessary” is the correct word here. Maybe it could be “interesting and novel”. But I leave this to the authors to decide.

Reply:

Yes, "interesting and novel" is a better expression in this sentence.

Comments:

Line 210: “KAUFMANN et al. (KAUFMANN et al., 1976)” should be “Kaufmann et al. (1976)”.

Reply:

Thanks for your comments. The sentence was amended in the revised paper.

Comments:

 Line 328: I think that “it weakly influenced by ...” should be “It is weakly influenced by ...”

Reply:

Thank you for your comments. We rechecked the article and corrected some mistakes of English expression.

Comments:

The reference list is arranged according to number, which are not used in the text. You should choose to use numbers (and in that case the references in the manuscript should be corrected according to this, or otherwise delete the numbers of each reference in the reference list and arrange them alphabetically. In fact, you should check the journals requirements for this.

Reply:

Thank you for your comments. We have reviewed the references and revised them.

Round 2

Author Response

Comments:

Equation (3) (incorrectly duplicated as Eq (4) in the revised version) and the preceding relationship between the variance contribution rate Ci and the corresponding eigenvalues arstill not correct. The variance contribution rate Ci and the corresponding eigenvalue λi should have the same subscript. The symbols used for the subscripts in the revised version of the denominator in Equation (3) have been changed from n to m in Eq. (4), but the algorithm represented by the equation is still wrong. The denominator of Equation (3) should be the sum of all n of the eigenvalues i.e. the sum of λj over the range j=1 to j=n. Lines 164 to 170 in the manuscript should be replaced with the following:

“The proportion of each eigenvalue λi in the sum of n eigenvalues is the variance contribution rate Ci which reflects the proportion of a spatial typical field representing the total features. The larger the variance contribution rate, the greater the weight of the spatial typical field. The formula for calculating the variance contribution rate is as follows:

Reply:

Thank you for your comments. We revised the formula in the revised version of the paper.

Comments:

Lines 351 and 353: Replace “Nmf2” with “NmF2”

Reply:

Thank you for your comments. Accepted and changed.

Comments:

Line 129: The reference [27] is inappropriate since it discusses version 1.94.2 of the TIE-CGM model of which the specifications are as follows (cut from reference [27], p. 75):

This is indeed inconsistent with the following claims in the revised manuscript (Lines 127 to 129):

“In this work, we use TIEGCM V2.0, which has a horizontal resolution of 127 2.5°×2.5° in geographic latitude and longitude, 57 pressure surfaces from ~97 km to 128 ~500 km, with a vertical resolution of ¼ scale height [27].”

The authors need to replace reference [27] with one that gives the claimed details of TIEGCM V2.0.

Reply:

Thanks for your comments. Accepted and changed to the references (Li et al., 2022).

Reviewer 2 Report

The author has anwered all my questions and concerns, and the paper now is improved much. However, it is still not ok for publishing. I give comments based on the lines of the track changes provided by authors (with blue lines)

Line 118-119 revise into 'simulate many major parameters in the thermosphere and ionosphere such as neutral temperature, wind and total electron content'

Line 122 here the solar radiation shall cite the related papers

Also, the  ionization by auroral precipitation shall be provided, the author shall check the recent published papers using tiegcm

Line 182-183 Fig 1 shows the geomagnetic indices in 2002 and 2008

Line 184 Dst

Line 186-187 give more detail on this, for example, how F10.7 variy in 2002 and 2008, with a maximum of XX? How geomagnetic activity is more active in 2002??

For section 4,3 to 4.4, the author shall clearly present the physical meanings of these second and third modes. Now it is pretty unclear whether these coefficient stand for some real physical meanings or not. 

Author Response

Comments:

Line 118-119 revise into 'simulate many major parameters in the thermosphere and ionosphere such as neutral temperature, wind and total electron content'

Line 182-183 Fig 1 shows the geomagnetic indices in 2002 and 2008

Line 184 Dst

Reply:

Thank you for your comments. We revised the questions you mentioned in the article. Figure 1 contains not only the geomagnetic index but also the F10.7 index, so it cannot be represented only by the geomagnetic indices.

Comments:

Line 122 here the solar radiation shall cite the related papers

Also, the ionization by auroral precipitation shall be provided, the author shall check the recent published papers using tiegcm

Reply:

Thanks for your comments. This paper focuses on using the TEC data of the SAA in 2002 and 2008 obtained from TIEGCM simulations to obtain the first three EOFs compare and analyze the spatial and temporal variation of TEC in the SAA in solar maximum and solar minimum. In addition, correlation analysis is performed using the annual coefficients to identify the possible drivers of TEC changes (F10.7, Dst, Ap), which did not involve the related content of auroral precipitation. We have carefully considered your suggestion, but we think it is more appropriate not to provide the content of auroral precipitation. Meanwhile, your suggestion can also be the direction of our further research. According to your suggestion, we have added several references. (Li et al., 2022; Kopp et al., 2016)

Comments:

Line 186-187 give more detail on this, for example, how F10.7 variy in 2002 and 2008, with a maximum of XX? How geomagnetic activity is more active in 2002??

Reply:

Thanks for your comments, we added more details to Figure 1 in the results section.

Comments:

For section 4,3 to 4.4, the author shall clearly present the physical meanings of these second and third modes. Now it is pretty unclear whether these coefficient stand for some real physical meanings or not. 

Reply:

Thank you for your comments. In Section 4.2, we introduced the first mode, including the first mode EOF and its time coefficient T1 (Figure 2c,2d,3a,3b). The second and third modes have the same physical meaning as the first mode. Therefore, its physical meaning is not emphasized in sections 4.3 and 4.4, but there may be some misunderstanding in the description. Therefore, we improved the expression of the second and third modes in section 4.3 and 4.4.

Reviewer 3 Report

The authors have answered all my comments and also included some of them in the revised version of their work.

I consider it can be published in its present form now.

Author Response

Comments:

The authors have answered all my comments and also included some of them in the revised version of their work.

I consider it can be published in its present form now.

Reply:

Thank you for your comments, so that our paper has been improved to the level of publication. Thank you again for your recognition of our paper.